# Inoculation with Arbuscular Mycorrhizal Fungi Alleviates the Adverse Effects of High Temperature in Soybean

**DOI:** 10.3390/plants11172210

**Published:** 2022-08-25

**Authors:** Kanchan Jumrani, Virender Singh Bhatia, Sunita Kataria, Saud A. Alamri, Manzer H. Siddiqui, Anshu Rastogi

**Affiliations:** 1Division of Plant Physiology, Indian Institute of Soybean Research, Indore 452001, India; 2School of Biochemistry, Devi Ahilya Vishwavidyalaya, Indore 452001, India; 3Department of Botany and Microbiology, College of Science, King Saud University, Riyadh 11451, Saudi Arabia; 4Laboratory of Bioclimatology, Department of Ecology and Environmental Protection, Faculty of Environmental Engineering and Mechanical Engineering, Poznan University of Life Sciences, Piątkowska 94, 60-649 Poznan, Poland

**Keywords:** arbuscular mycorrhizal fungi, biofertilizer, chlorophyll fluorescence, high temperature, photosynthesis, soybean

## Abstract

High temperature is foremost abiotic stress and there are inadequate studies explicating its impact on soybean. In this study, a pot experiment was done in a greenhouse maintained at a day/night temperature of 42/28 °C with a mean temperature of 35 °C to examine the effects of high temperature in soybean plants inoculated with and without arbuscular mycorrhizal fungi (AMF).Various parameters were taken in soybean plants treated with AMF (+) and AMF (−) such as growth analysis, chlorophyll content, canopy temperature, number of stomata, gas exchange, chlorophyll fluorescence, seed yield, and its attributes. It was observed that growth parameters like leaf area, stem height, root length, shoot and root dry biomass were increased in AMF (+) as compared to AMF (−) plants. Chlorophyll content, the number of stomata, photosynthesis rate, stomatal conductance, transpiration rate, and water use efficiency increased in AMF (+) as compared to AMF (−) plants. Chlorophyll fluorescence parameters such as Fv/Fm, Fv/Fo, PhiPSII, fluorescence area, performance index, photochemical quenching, linear electron transport rate, and active reaction centres density of PSII were also found to be enhanced in AMF (+) plants. However, canopy temperature, intercellular CO_2_, Fo/Fm, and non-photochemical quenching were higher in AMF (−) as compared to inoculated plants. An increase in growth and photosynthesis ultimately enhanced the seed yield and its attributes in AMF (+) as compared to AMF (−). Thus, AMF (+) plants have shown much better plant growth, photosynthesis parameters, and seed yield as compared to AMF (−) plants under high temperature. Thus, it is concluded that heat stress-induced damage to the structure and function of the photosynthetic apparatus was alleviated by AMF inoculum. Therefore, AMF can be used as a biofertilizer in alleviating the adverse effects of heat stress in soybean.

## 1. Introduction

High-temperature stress is considered as one of the major destructive stress among various abiotic stresses. Heat stress is defined as the increase in temperature above a threshold level to cause irretrievable damage to plants [1,2]. The intensity and impact of high temperature are predicted to increase more with long-lasting effects in the coming years [3]. High temperature has damaging ramifications on plants’ morphological, physiological, and biochemical processes, which eventually leads to the down-regulation of plants’ productivity [4]. Soybean is the world’s most important seed legume [5,6,7]. Since the introduction and inception of commercial cultivation in India in the late ’60s, the crop is being cultivated in around 11.8 million hectares with a production of 13.5 million tons (2020–2021 estimates). Presently soybean is contributing 42% share of total oilseed and 22% to total oil production in the country. With an increase in population, the demand for edible oil is increasing and 40% of the demand is being fulfilled by different oil seed crops and the rest 60% of demand is being made up by imports to India. The cost of the import of edible oil put high pressure on our foreign exchange. Among all the oilseed crops, soybean is having the highest potential to meet the challenge of being self-sufficient in the production of edible oil. Soybean is frequently subjected to high-temperature stress, due to which its growth, seed yield, and quality are greatly affected [8,9,10]. The occurrence of temperature above a threshold level particularly beyond 35 °C has often been observed in many regions of India where soybean is grown [11]. Thus, any further rise in temperature may brutally obstruct soybean productivity as the current temperatures in major soybean growing regions in India are already at the threshold of the upper limit.

Plants have a number of tolerance mechanisms for forestalling the harmful effects of abiotic stress conditions. Among the environmentally safe sustainable endeavors, the association of Arbuscular mycorrhizal fungus (AMF) with the roots of the plant can be reconnoitered to improve plant growth and productivity under abiotic stress. Thus, the presence of AMF improves plants’ tolerance to high temperatures when compared to the plants not colonized with AMF [12]. AMF being a potential biofertilizer alleviates the harmful effect of high temperature on the plant by enhancing plant growth and yield [13,14]. AMF is obligate biotrophs that can form symbiotic associations with the roots of plants [15], which plays a key role in increasing crop productivity and nutrient cycle [16]. Also, the use of AMF can reduce the use of inorganic fertilizers and pesticides in agriculture due to bio-protection.

AMF usually form a mutualistic symbiosis with many plants to control the host plants’ growth under abiotic stress conditions by various direct or indirect mechanisms. The fungus in such a relationship forms a symbiotic association with the roots of the plant and enhances its ability to absorb water and nutrition by increasing mycelium area [17],in turn, the plant provides carbohydrates to the fungal partner [18,19,20,21,22]. AMF symbiosis alleviates temperature stress to the host plants via improving uptake of nutrients [23] and enhances the photosynthetic rate and photochemistry of PSII [24,25,26,27], improves the osmotic adjustment [28], superior antioxidant activity [29,30], and better reproductive capacity [31]. Many studies have been conducted on AMF and abiotic stress, but all the studies have been conducted in controlled conditions in sterilized soil. Under sterilized soil, actual field conditions cannot be mimicked therefore in this study unsterilized or natural soil was used for the experiments. AMF is likely to be more important in a future sustainable agriculture system. Till now, many reports are there on heat stress and AMF separately but to the best of our knowledge, this is the first elaborative report about alterations in gas exchange parameters and photosynthetic apparatus in response to heat stress in the presence of AMF in soybean. Hence, the main aim of this study was to inspect the role of AMF in assuaging the harmful effects of high-temperature stress in soybean plants. Thus, this study could contribute toward the understanding of plant tolerance to high-temperature stress in soybean.

## 2. Results

Various growth parameters (such as leaf area, plant height, root length, above and below-ground biomass), canopy temperature, photosynthetic efficiency measurements such as chlorophyll a fluorescence parameters, net photosynthesis rate, transpiration rate, stomatal conductance, water use efficiency, stomatal density, total chlorophyll content, seed yield, and its attributes were determined in AMF (+) and AMF (−) soybean plants.

### 2.1. Effect on Growth Parameters and Root Colonization

Leaf area, plant height, root length, above and below-ground biomass were measured for AMF (+) and AMF (−) plants. The leaf area of AMF (+) plants was 30% more as compared to AMF (−) plants. In AMF (+) plants, plant height was increased by 34% and root length was increased by 41% as compared to AMF (−) plants. In AMF (+) plants the shoot dry weight was 20% and root dry weight was 44% more as compared to AMF (−) plants (Figure 1). The nodules dry weight of AMF (+) plants was 33% more as compared to AMF (−) plants. In AMF-treated plants, root colonization was found to be ~50–60% including the vesicles and arbuscules, while very less colonization was observed in control plants (~5–7%).

### 2.2. Effect on SPAD Value, Canopy Temperature and Stomatal Density

Total Chlorophyll content was more in AMF (+) soybean plants as compared to AMF (−) plants. AMF (+) soybean plants had 24% more chlorophyll content as compared to AMF (−) plants. Canopy temperature was also measured in AMF (+) and AMF (−) plants. It was evident from the result that AMF (+) plants (32.0 °C) had lower canopy temperatures as compared to AMF (−) plants (37.4 °C). Leaves of plants inoculated with AMF showed significantly higher stomatal frequency. The number of stomata in AMF (+) soybean plants was 318/mm^2^ while it was 233/mm^2^ in AMF (−) plants (Figure 2).

### 2.3. Effect on Gas Exchange Parameters

Photosynthetic parameters were studied in AMF (+) and AMF (−) plants under high temperatures. The rate of photosynthesis was enhanced by 32% in AMF (+) plants as compared to AMF (−) plants under high-temperature conditions. Transpiration was increased by 27%, stomatal conductance by 20%, and water use efficiency by 22% in AMF (+) plants while intercellular CO_2_ was low (11%) in AMF (+) plants as compared to control plants. It indicated that AMF (+) plants had higher photosynthetic efficiency as compared to AMF (−) plants (Figure 3).

### 2.4. Effect on Chlorophyll a Fluorescence

Different chlorophyll fluorescence parameters were studied in AMF (+) as well as AMF (−) plants under high-temperature conditions. Electron transport rate, photochemical quenching, Fv/Fm, and PhiPSII enhanced under high temperature in AMF (+) plants while non-photochemical quenching decreased. Electron transport rate was increased by 30%, Fv/Fm ratio by 17%, PhiPSII by 30%, and photochemical quenching by 35% in AMF (+) plants while non photochemical quenching declined by 20% as compared to AMF (−) plants (Figure 4).

### 2.5. Effect on Energy Pipeline Leaf Model

The energy pipeline leaf model was inferred using biolyzer HP3 software (Figure 5). This model gives evidence about the efficacy of the flow of energy from antennae to the electron transport chain components through the reaction centre of PSII. Energy flow was calculated from the pipeline leaf model for AMF (−) and AMF (+) plants. The flow of energy, photons trapped and used for efficient primary photochemistry or in electron transport system was studied by the width of the arrows (Figure 5). A phenomenological leaf model depicts more active reaction centres per unit area of the AMF (+) plants as compared to AMF (−). In this model, open circles represent the active reaction centre and AMF (+) plants had more active reaction centres and higher electron transport efficiency as shown by the broader width of the arrow in the leaf models as compared to AMF (−) plants under high temperature (Figure 5). The following effects were observed in soybean leaves (phenomenological per excited cross-section (CS) area): ABS/CSo, ETo/CSo, DIo/CSo, and TRo/ CSo decreased in AMF (−). ABS/CSo describes the number of photons absorbed by antenna molecules of active and inactive PSII RCs over the excited cross-section and is represented by the dark-adapted Fo. The ABS/CSo can be substituted as an approximation by the fluorescence intensity, Fo. A decrease in ABS/CSo at high temperature indicates a decrease in the energy absorbed per excited cross-section. ETo/CSo represents electron transport in a PSII cross-section and indicates the rate of reoxidation of reduced Q_A_ via electron transport over a cross-section of active RCs. A decrease in this ratio indicates the inactivation of RC complexes and the OEC and also suggests that the donor side of PSII has been affected. DIo/CSo represents the total dissipation measured over the cross-section of the leaf that contains active and inactive RCs. TRo/CSo represents the maximal rate by which an exciton is trapped by the RC resulting in the reduction of Q_A_. The ratio TRo/CSo also decreased in AMF (−) leaves indicating low energy trapping by reaction centres. A decrease in the density of active RCs (indicated as open circles) and an increase in the density of inactive RCs (indicated as filled circles) were observed in response to AMF (−) (Figure 5).The area over the fluorescence induction was increased by 28% in AMF (+) plants as compared to AMF (−) plants. The performance index was increased by 33% and Fv/Fo was increased by 21% in AMF (+) plants as compared to AMF (−) plants while Fo/Fm was decreased by 27% in AMF (+) plants as compared to AMF (−) plants (Figure 6).

### 2.6. Effect on Seed Yield and Its Attributes

Seed yield and its attributes were increased in AMF (+) as compared to control plants. Seed yield was 12.1 g/plant in AMF (−) plants which was increased by 33% in AMF (+) plants (16.1 g/plant). Pod number, seed number, and total biomass were also increased in AMF (+) plants as compared to AMF (−) plants. The number of pods was increased by 33%, the number of seeds by 37%, and total biomass by 32% in AMF (+) plants as compared to AMF (−) plants under high-temperature stress (Figure 7).

## 3. Discussion

Among the changing mechanisms of the environment, the continuously increasing temperature is considered one of the utmost harmful stress [32]. AMF forms a symbiotic association with higher plant roots by promoting plant growth and nutrient uptake, which is of biological significance [15,33,34,35]. The present study was piloted in a greenhouse, to examine the effects of AMF on growth, chlorophyll content, canopy temperature, number of stomata, gas exchange, chlorophyll fluorescence, seed yield, and its attributes when soybean plants were grown under high-temperature stress. The present study clearly showed that the soybean plants infected with AMF showed better growth characteristics (leaf area, dry weights, and length of shoot and roots) than non-AMF plants under high-temperature stress (35 °C). This result was in agreement with earlier reports which have also shown the positive effects of AMF on plant growth [25,29,36,37,38,39,40]. AMF symbiosis also increases leaf number, and leaf area and delays senescence [41]. These affirmative results may be due to the improvement of gas exchange parameters [20,34] and an increase in root density [42,43,44]. Nodule dry weight was also enhanced in plants inoculated with AMF. Studies have shown that AMF effects are also associated with root nodules [45,46]. This might point to interactions between AMF and rhizobia inside the root nodules, the root organs in which nitrogen fixation takes place. Moreover, molecular identification revealed that under natural circumstances, root nodules are colonized by specific AMF communities that are similar in different legume species and different from the root AMF communities [47]. The presence of AMF in the nodules could indicate that AMF deliver nutrients that are essential for nitrogen fixation directly into the nodules.

Photosynthesis has been considered an important indicator of growth because of its direct association with the productivity of crops [48,49,50]. High temperature could affect photosynthesis by stomatal or non-stomatal factors [51,52,53]. The present study data showed a significant increase in the net photosynthetic rate, transpiration rate, and stomatal conductance in AMF plants compared to non-AMF plants, under high temperatures. The water use efficiency was also high in AMF plants as compared to non-AMF plants. It clearly indicated that AMF can provide enhanced gas exchange ability by increasing CO_2_ assimilation and transpiration and decreasing stomatal resistances which are necessary to supply photosynthates needed by fungal symbionts [26,34,54,55,56]. Previous reports have also shown that higher rates of stomatal conductance in AMF plants improve the demand for transpiration in relation to non-AMF plants under high-temperature stress [26,57]. It proves that inoculation by mycorrhizal fungi improves the photosynthesis of AMF-treated plants under high-temperature conditions.

Chlorophyll fluorescence parameters are useful for environmental studies in assessing plant responses to eco-physiological adversities [58,59,60,61,62]. As compared to non-AMF under high-temperature plants, AMF plants were able to convert light energy more efficiently and reduced the damage to photosynthetic apparatus [63]. The present study showed a significant increase in the performance index, Fv/Fm, photochemical quenching, PhiPSII, electron transport rate, and Fv/Fo in AMF (+) plants compared to non-AMF plants. Under high-temperature AMF appears to have protected the water-splitting complex and subsequently better primary photochemistry of PSII. The effect of high temperature clearly led to reduced performance index values, and area for AMF (−) plants, whereas AMF (+) plants maintained higher performance index levels. Application of AMF caused higher Fv/Fm in the mycorrhizal colonized plants as compared with AMF (−) plants, which is in accordance with the results of Ruiz-Lozano et al. [64].

In addition, inoculation improved photosynthesis by inducing changes in leaf anatomical characters. The more number of stomata in AMF-treated plants could be construed as a plant stratagem to fulfill the higher demand for CO_2_ needed to match the higher growth rate in plants inoculated with arbuscular mycorrhizal fungi [65,66,67]. AMF inoculated soybean plants showed lower canopy temperature under high temperatures as compared to non-inoculated plants, indicating AMF (+) plants were able to use more of the available water in the soil to keep their canopy cooler. The AMF plant leaves had higher chlorophyll content compared to non-AMF plants, this greater chlorophyll content means it subsequently led to higher photosynthetic efficiency and biomass production under heat stress [26,68,69]. Enhancement in biomass and photosynthesis ultimately helped the AMF (+) plants to produce more seed yield under high-temperature conditions as compared to AMF (−) plants. Thus, this study provides the role of AMF in alleviating the adverse effects of high-temperature stress in soybean plants.

## 4. Materials and Methods

### 4.1. Experimental Design

The experiment was performed under greenhouse conditions at ICAR-Indian Institute of Soybean Research, Indore (22.72° N, 75.83° E). The most popular soybean genotype JS 20–29 was grown in cement pots (45 cm height and 18 cm diameter) with 8 replicates in randomized complete block design for each AMF (+) treated pot and without AMF (−) pots maintained at day/night temperatures of 42/28 °C with a mean temperature of 35 °C. The pots were soaked with tap water 24 h before planting. Before sowing, the seeds were treated with recommended fungicides, viz. bavistin, and dithane M, and then inoculated with slurry of *Rhizobium japonicum*. Four seeds of uniform size were sown at 4 cm depth in each pot. One week after sowing, thinning was done to two plants per pot, which were maintained until maturity. The plants were irrigated daily to avoid water stress and appropriate measures were taken to keep the plants free from any biotic stresses.

### 4.2. AM Fungi Inoculum

AM fungal culture was procured from the Microbiology Section, Indian Institute of Soybean Research, and used for the experiment. Microcosm trial was conducted in Vertisols soil (pH 7.5) (1:2.5 soil water ratio) carbon (0.5%); P (6.2 mg/kg); N (6.4 mg/kg)] with and without AMF inoculum. For AMF treatment culture, 20 g of crude soil was inoclutated with AMF inoculum (1000 spores of dominant populations of *Rhizophagus irregularis*, *Funneliformis mosseae*, and *Funneliformis geosporum*) consisting of propagules, vesicles, hyphal fragments, and infected root pieces per pot by layering method just below the seeds, completely intermixed in the potting mixture whereas, no starter culture was added in the non-AMF set of experiments. Plants without AMF inoculation were the control plants while AMF (+) were the AMF inoculated ones.

### 4.3. Growth Parameters and Assessment of AMF Root Colonization

All the observations were taken at the R5 stage (initiation of seed fill) as at this stage the plant attains its maximum height, node number, nutrient accumulation, leaf area, dry weight and photosynthesis For analyzing the growth in terms of leaf area and dry matter (above and below-ground biomass), plants from three pots were sampled at the R5 stage from all the plants of AMF (+) and AMF (−) pots. The plant parts were separated into leaves, stem, pod, and root, was oven-dried at 70 °C, and then dry weights were recorded. Plant height, root length and leaf area were also measured in these plants. The leaf area was recorded using a leaf area meter (Model LI-3100, LICOR Nebaraska, Lincoln, NE, USA). Mycorrhizal colonization in roots was examined under microscope (LEADZ optics model LOS 400). Mycorrhizal colonization in fresh roots was determined after digesting and clearing the roots in trypan blue acid (0.05%) in lactoglycerol by method of Biermann and Linderman [70]. Then roots were washed under running tap water and then cut into 1 cm pieces. Roots were cleared with 10% KOH solution at 90–95 °C water bath for 1 h. Roots were treated with alkaline H_2_O_2_ for 10 min, washed thoroughly with tap water and acidified with 1% HCl solution for 2 min and stained in 0.05% trypan blue with lactoglycerol.

### 4.4. Canopy Temperature

Canopy temperature was recorded at the R5 stage in three replications (5 plants/replication) using a non-contact infrared thermometer (Palmer Wahl, model DHS115XEL) [71]. The thermometer was held in such a manner that the sensor viewed only the canopy of the plant and prevented from sensing the soil surface. All the measurements of canopy temperature were made in three replications (5 plants/replication) in all the plants of AMF (+) and AMF (−) pots in the late morning to early afternoon cloudless period (11.00–15.00 h).

### 4.5. Chlorophyll Content

Non-damaging calculation of chlorophyll content was done using a portable Minolta chlorophyll meter (SPAD-502, Spectrum Technologies, Inc., Plainfield, IL, U.S.) which is based on leaf transmittance at 650 and 940 nm [72]. The readings were recorded at the R5 stage on a fully expanded third leaf from the top in all the plants of AMF (+) and AMF (−) pots in three replications (5 plants/replication).

### 4.6. Stomatal Density

To measure the stomatal density, the impression approach was used and is expressed as the number of stomata per unit leaf area [73]. The leaves selected were those for which gas exchange parameters were measured. Leaves were cleaned with cotton and carefully smeared with the transparent nail varnish in the mid-area between the central vein and the leaf edge for ~20 min. After drying, the thin film was peeled off from the leaf surface, mounted on a glass slide, immediately covered with a cover slip, and then lightly pressured with fine-point tweezers, and a number of stomata for each filmstrip was counted under a microscope (model LOS 400, LEADZ Optics, London, UK).

### 4.7. GasExchange Measurements

Gas exchange parameters like net photosynthetic rate, stomatal conductance, transpiration rate, water use efficiency, and intercellular CO_2_ were measured at the R5 stage in three replications (5 plants/replication) in fully expanded leaf from the top in all the plants of AMF (+) and AMF (−) pots using photosynthesis system (LI-6400 XT, LI-COR Inc., Lincoln, NE, USA). Gas exchange measurements were made at PAR of 1000 µmol m^−2^s^−1^ and ambient CO_2_ concentration (390–400 μmol CO_2_ mol^−1^air) between 10.00–12.00 h.

### 4.8. Chlorophyll a Fluorescence

Chlorophyll fluorescence parameters were recorded in the dark-adapted leaf (30 min) in all the plants of AMF (+) and AMF (−) pots in three replications (5 plants/replication) using 6400–40 leaf chamber fluorometer combined with LI-6400 XT. The fluorometer can be used to take measurements on both dark and light-adapted leaves. Measured parameters include Fo, Fm, Fs, Fo’, and Fm’ and calculated parameters include Fv/Fm, PhiPSII, electron transport rate, photochemical (qP), and non-photochemical quenching (qN).

### 4.9. Energy Pipeline Leaf Model

Chlorophyll fluorescence induction kinetics were recorded in the dark-adapted leaf (30 min) in all the plants of AMF (+) and AMF (−) pots in three replications (5 plants /replication) using a Handy PEA Fluorimeter (Plant Efficiency Analyzer, Hansatech Instruments King’s Lynn, Norfolk, UK) from 10.00–12.00 h. The energy pipeline model was prepared using chlorophyll fluorescence; the data acquisition was for every 10 ms for the first 2 ms and every 1 ms thereafter by analyzing data in the program Biolyzer HP 3 software. The area over the fluorescence induction curve between Fo and Fm is presented by area. Each relative value is represented by the size of the proper parameters (arrow), the width of each arrow denotes the relative size of the fluxes or the antenna, empty circles represent reducing QA reaction centres (active), and full black circles represent non-reducing QA reaction centre(inactive or silent). The area of the arrows for each of the parameters, ABS/CSo, TRo/CSo, ETo/CSo and DIo/CSo, indicate the efficiency of light absorption, trapping, electron transport and dissipation per cross-section of PSII, respectively.

### 4.10. Seed Yield and Its Attributes

At harvest maturity, plants from the remaining five pots from each treatment were sampled and data on seed weight, pod number, seed number, and total dry weight per plant was recorded.

### 4.11. Statistical Analysis

Analysis of variance was carried out for each treatment and all the data sets using SAS statistical software (ver. 9.2).

## 5. Conclusions

We put forward the evidence that under high-temperature exposure AMF enrichment can facilitate high photosynthetic capacity and prevent photosynthetic apparatus from being damaged. The information gained from this study would be beneficial for perilously evaluating the potential role of AMF in alleviating the adverse effects of high-temperature stress. Beneficial microorganisms, for instance, AM fungi are attractive to farmers in the milieu of sustainable agriculture. This finding suggested that the inoculation of AMF might be a justifiable method for enhancing plant growth performance under adverse environmental conditions. The present study clearly defines the function of AMF in improving the high-temperature tolerance of soybean plants by enhancing chlorophyll content, photosynthetic rate, stomatal conductance, WUE, linear electron transport, and seed yield. Results of the present study show the possible role of mycorrhizal inoculation as a bio-fertilizer is strongly supported and its ability to enhance soybean productivity under high-temperature stress conditions. These studies conclude that AMF inoculation can alleviate the damaging effects of high-temperature stress on the growth and yield of soybean. There is no doubt that the growth of many plants can be substantially improved if they possess a well-developed mycorrhizal system. There is an urgent need for understanding this mechanism and to design an appropriate plant–AMF combination for better use of natural resources. Future research, therefore, should be based on holistic approaches involving multidisciplinary sciences, such as plant and fungal physiology, and soil and molecular biology, for a better understanding of the processes in the plant–AMF–soil continuum. Thus, it is concluded that heat stress-induced damage to the structure and function of the photosynthetic apparatus was alleviated by AMF colonization.

## Figures and Tables

**Figure 1 plants-11-02210-f001:**
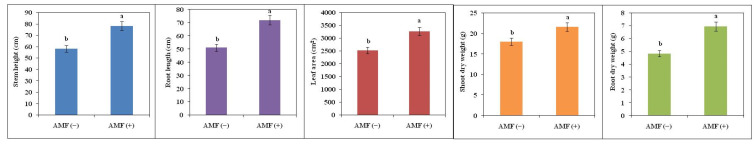
Plant height, root length, leaf area, shoot dry weight, and root dry weight of non-inoculated AMF (−) and inoculated plants AMF (+) of soybean. The vertical bar indicates ± SE for the mean. The means for each main treatment followed by the different letters are significantly different (*p* ≤ 0.05).

**Figure 2 plants-11-02210-f002:**
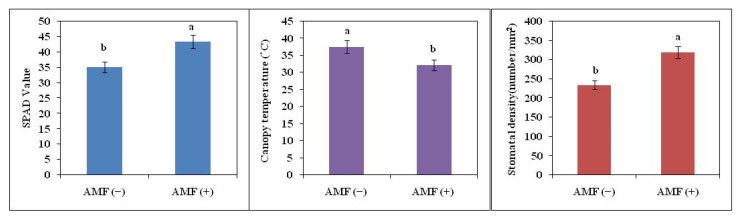
Chlorophyll content (SPAD value), canopy temperature, and the number of stomata on the abaxial surface of non-inoculated AMF (−) and inoculated plants AMF (+) of soybean. The vertical bar indicates ± SE for the mean. The means for each main treatment followed by the different letters are significantly different (*p* ≤ 0.05).

**Figure 3 plants-11-02210-f003:**
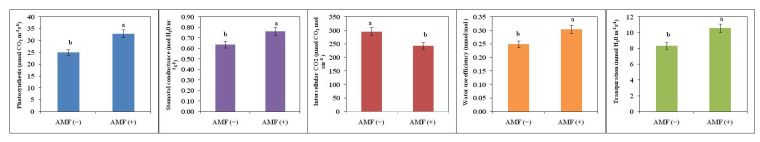
Rate of photosynthesis, stomatal conductance, intercellular CO_2_, water use efficiency, and rate of transpiration of non-inoculated AMF (−) and inoculated plants AMF (+) of soybean. The vertical bar indicates ± SE for the mean. The means for each main treatment followed by the different letters are significantly different (*p* ≤ 0.05).

**Figure 4 plants-11-02210-f004:**
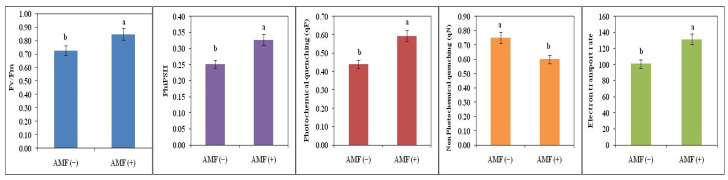
Maximum PSII efficiency (Fv/Fm), PhiPSII, photochemical quenching, non-photochemical quenching, electron transport rate of non-inoculated AMF (−) and inoculated plants AMF (+) of soybean. The vertical bar indicates ± SE for the mean. The means for each main treatment followed by the different letters are significantly different (*p* ≤ 0.05).

**Figure 5 plants-11-02210-f005:**
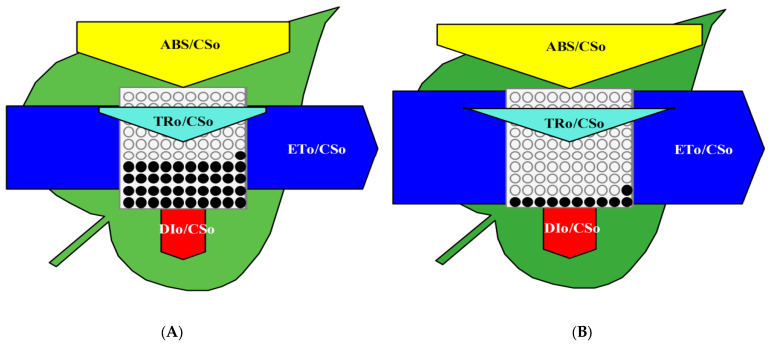
Phenomenological leaf model of leaves from non-inoculated (**A**) AMF (−) and inoculated plants (**B**) AMF (+) of soybean. Each relative value is represented by the size of the proper parameters (arrow), empty circles represent reducing QA reaction centres (active), and full black circles represent non-reducing QA reaction centres (inactive or silent). Efficiency of light absorption (ABS/CS_O_), trapping (TR_O_/CS_O_), and electron transport (ETo/CS_O_) and dissipation per cross-section of PSII (DI_O_/CS_O_) (phenomenologically per excited cross-section (CS) area).

**Figure 6 plants-11-02210-f006:**
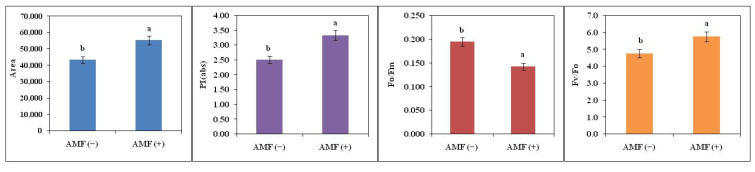
Fluorescence area, performance index (PI), Fo/Fm and Fv/Fo of non-inoculated AMF (−) and inoculated plants AMF (+) of soybean. The vertical bar indicates ± SE for the mean. The means for each main treatment followed by the different letters are significantly different (*p* ≤ 0.05).

**Figure 7 plants-11-02210-f007:**
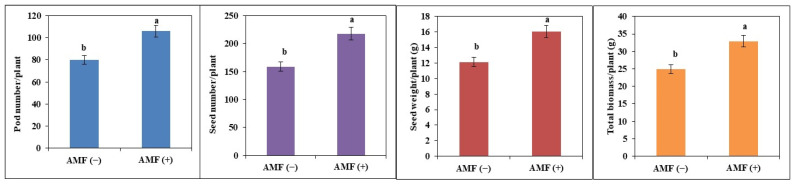
Pod number, seed number, seed weight, and total biomass of non-inoculated AMF (−) and inoculated plants AMF (+) of soybean. The vertical bar indicates ± SE for the mean. The means for each main treatment followed by the different letters are significantly different (*p* ≤ 0.05).

## Data Availability

Data can be obtained on request from the authors.

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
