# Peer review of "Inoculation with Arbuscular Mycorrhizal Fungi Alleviates the Adverse Effects of High Temperature in Soybean"

_plants, 2022, doi:10.3390/plants11172210_

Round 1

Reviewer 1 Report

The manuscript reports that arbuscular mycorrhizal fungi alleviates harmful effects of high-temperature stress in soybean. The results are interesting and will be of interest to other researchers. I have a few suggestions for a revision to improve the clarity of the study for the reader.

Line 17: the expression of AMF (+) and AMF (-) were different in the abstract and other aspects.

Line 116: internal should be intercellular.

Line 172/197/202: semicolon should be comma

Line 235: was the soil sterilized

Figure 5: The A/B are not present in this figure. You also should give more information to explain this figure.

Figure 6: Should PI be performance index in this figure? And also leaf area.

Author Response

Specific comments

The manuscript reports that an arbuscular mycorrhizal fungus alleviates harmful effects of high-temperature stress in soybean. The results are interesting and will be of interest to other researchers. I have a few suggestions for a revision to improve the clarity of the study for the reader.

Answer:  We thank the reviewer for providing useful comments/suggestions that have helped us to further improve this manuscript. Following are the responses to the comments and suggestions of the reviewer.

Comment-1 Line 17: the expression of AMF (+) and AMF (-) were different in the abstract and other aspects.

Answer: The expression of AMF (+) and AMF (-) has been corrected and the same in the whole manuscript.

Comment-2 Line 116: internal should be intercellular.

Answer: internal has been changed to intercellular.

Comment-3 Line 172/197/202: semicolon should be comma

Answer: semicolon has been replaced by a comma.

Comment-4 Line 235: was the soil sterilized

Answer: No, the soil was not sterilized; soil from the long-term soybean-based farming system managed at ICAR-Indian Institute of Soybean Research (IISR) Indore; was collected. Many studies have been conducted on AMF and abiotic stress, but all the studies have been conducted in controlled conditions in sterilized soil. Under sterilized soil actual field conditions cannot be mimicked,  therefore in this study unsterilized or natural soil was used for the experiments.

Comment-5 Figure 5: The A/B are not present in this figure. You also should give more information to explain this figure.

Answer: The A/B has been added to the figure. The figure legends have been modified.

Comment-6 Figure 6: Should PI be the performance index in this figure? And also leaf area.

Answer: Yes, PI is the performance index in this figure. It is not the leaf area it is the area over the fluorescence induction curve between Fo and Fm.

Reviewer 2 Report

The study presented in this manuscript was designed to assess the effects of arbuscular mycorrhizal fungi on tolerance of heat stress in a soybean cultivar. Given the increasing attention to the detrimental effects of unusually high temperatures on agriculture, coupled with increasing recognition of the fundamental role of plant microbes on plant health and growth, this study has the potential to be of broad interest. The results are relatively straightforward, and present a compelling story. Unfortunately, there are a couple of areas where the nature of the study were quite unclear to me, making it difficult to assess the robustness of the results. Also, the conclusion drawn from the study - that AM fungi will offset heat stress in soybeans - seems to be overly broad, given this is a single study under a specific set of conditions.

Most of my concerns were with the description of the methods. For example, I couldn’t find an indication of the watering regime, which is crucial to assessing the potential relevance of this study to understanding the effects of heat stress on soybeans.

This issue, of not enough information about the methods, applied across the methods section, which could use a substantial rewrite to present the methods in a way that the study could be repeated by other researchers.

Also, it appears plant N and P contents weren’t measured; this info also is critical for interpreting the results, given changes in N and P status may play key roles in alleviating heat stress.

And, I couldn’t find information about whether and how the soybeans were inoculated with nodule-forming bacteria, or if nodulation was measured in the study.

As a minor note, the formatting of the Y-axis labels in many of the graphs made them difficult to read. Also, the manuscript would benefit from rewriting to standardise the English. The train of thought was somewhat difficult to follow in some areas because I had difficulty understanding the writing.

Author Response

Specific comments

The study presented in this manuscript was designed to assess the effects of arbuscular mycorrhizal fungi on tolerance of heat stress in a soybean cultivar. Given the increasing attention to the detrimental effects of unusually high temperatures on agriculture, coupled with increasing recognition of the fundamental role of plant microbes on plant health and growth, this study has the potential to be of broad interest. The results are relatively straightforward and present a compelling story. Unfortunately, there are a couple of areas where the nature of the study was quite unclear to me, making it difficult to assess the robustness of the results. Also, the conclusion is drawn from the study - that AM fungi will offset heat stress in soybeans - seems to be overly broad, given this is a single study under a specific set of conditions.

Answer: We thank the reviewer for providing useful comments/suggestions that have helped us to further improve this manuscript. Following are the responses to the comments and suggestions of the reviewer. The conclusion has been improved

Comment-1 Most of my concerns were with the description of the methods. For example, I couldn’t find an indication of the watering regime, which is crucial to assessing the potential relevance of this study to understanding the effects of heat stress on soybeans.

Answer: The pots were soaked with tap water 24 hrs before planting. The plants were irrigated daily to avoid water stress and appropriate measures were taken to keep the plants free from any biotic stresses. The text related to it is now added to the MS.

Comment-2 This issue, of not enough information about the methods, applied across the methods section, which could use a substantial rewrite to present the methods in a way that the study could be repeated by other researchers.

Answer: Materials and methods section has been improved

Comment-2 Also, it appears plant N and P contents weren’t measured; this info also is critical for interpreting the results, given changes in N and P status may play key roles in alleviating heat stress.

Answer: Yes we do agree that N and P measurement may have helped this study to go into more mechanism part, but even without that this study has significance in understanding how the AMF may help the plant under stress but for future studies, the N and P contents will be measured and we will try to go deeper in its understanding.

Comment-3 And, I couldn’t find information about whether and how the soybeans were inoculated with nodule-forming bacteria, or if nodulation was measured in the study.

Answer: Information has been added to the revised text.
Before sowing, the seeds were treated with recommended fungicides and Rhizobium culture. Microcosm trial was conducted in Vertisols soil [pH 7.5 (1:2.5 soil water ratio); organic carbon (0.5%); Olsen P (6.2 mg/kg); mineral N (6.4 mg/kg)] with AMF and without AMF colonization. For AMF treated set, the starter culture of 50 g crude soil AM inoculum (infected root bits, mycelium and hyphae) was inoculated by layering method just below the seeds, completely intermixed in the potting mixture whereas, in the case of the non-AMF set (control), no starter culture was added. Sorry, the nodulation was not measured in this study.

Comment-4 As a minor note, the formatting of the Y-axis labels in many of the graphs made them difficult to read.

Answer: The formatting of the Y-axis labels in the graphs has been made clear now.

Comment-5 Also, the manuscript would benefit from rewriting to standardize the English. The train of thought was somewhat difficult to follow in some areas because I had difficulty understanding the writing.

Answer:  English has been improved in the manuscript.

Reviewer 3 Report

The present study investigated the effects of high temperature in soybean plants colonized with and without arbuscular mycorrhizal fungi (AMF). The language/grammar used in the Manuscript requires significant improvement. Several statements are lack clarity and the writing style is appalling and requires a major revision. Manuscript has no novelty, and little scientific value. Because there are many reports on the effect of AMF on plant growth and physiological properties under high temperature. It is important to indicate new findings of this study.

Comment

Titel: “high-temperature stress” “harmful effects” , high temperature has negative effect on plant physiology, thus modify title of Manuscript

Line 38: “Among the changing mechanisms of the environment,”, what is changing mechanism?

Lines 36-40, describing same thing, negative effect of high temperature on plant

Line 50-51: “..require cavernous understanding of soybean responses to heat stress”. This study aims not to investigate plant response to heat stress, but AMF inoculation. Thus justification should be revised

Results part

It is important to measure and maintain soil water holding capacity, because of high temperature. But this study doesnot describe about irrigation and soil condition.

Figure 5, picture doesnot show treatments

In discussion section most of references are outdated, need to include recent studies. Moreover, there are many reports on the mitigation of heat stress in plants by AMF and plant physiological changes. What is new findings in this study? Please describe about it as well.

Moreover, soybean is leguminous plant, why no symbiotic performance of plant with rhizobia were described. Also nutrient acquisition of plant, e.g. N,P,K contents in plant tissue should be investigated.

Conclusion is too general, what was the hypotheses of this study?, what is new in this study? The conclusion should solely support the objectives of the study.

Material and methods

Line 229:

The pot experiments and greenhouse conditions should be clearly described. From where soil samples were collected?, soil chemical physical characterisation, irrigation and soil moisture description.

Line 289: “total dry weight”, plant or seeds?

Author Response

Specific comments

The present study investigated the effects of high temperature in soybean plants colonized with and without arbuscular mycorrhizal fungi (AMF). The language/grammar used in the Manuscript requires significant improvement. Several statements are lack clarity and the writing style is appalling and requires a major revision. Manuscript has no novelty, and little scientific value. Because there are many reports on the effect of AMF on plant growth and physiological properties under high temperature. It is important to indicate new findings of this study.

Answer: We thank the reviewer for providing useful comments/suggestions that have helped us to further improve this manuscript. Following are the responses to the comments and suggestions of the reviewer. English has been improved in the manuscript. The study discusses the mechanism of AMF-induced protection of the photosynthetic apparatus of soybean against heat stress. Till now, many studies have been conducted on heat stress and AMF separately but to the best of our knowledge, this is the first elaborative report about alterations in gas exchange parameters and photosynthetic apparatus in response to heat stress in the presence of AMF for soybean plants.

Comment-1Title: “high-temperature stress” “harmful effects”, high temperature has negative effect on plant physiology, thus modify title of Manuscript.

Answer: Title has been modified accordingly, “Inoculation with arbuscular mycorrhizal fungi alleviates the adverse effects of high-temperature stress in soybean”.

Comment-2 Line 38: “Among the changing mechanisms of the environment,” what is changing mechanism?

Answer: Changing mechanisms are the environmental stresses (altered light, heat, drought, flooding, cold, salt, nutrient deficiencies, and heavy metals) which are important determinants of physiological plant processes.

Comment-3 Lines 36-40, describing the same thing, the negative effect of high temperature on plant

Answer: The sentence has been reframed.

Comment-4 Line 50-51: “..require a cavernous understanding of soybean responses to heat stress”. This study aims not to investigate plant response to heat stress, but to AMF inoculation. Thus justification should be revised

Answer: The sentence has been improved and reframed for more clarity.

Results part

Comment-5 It is important to measure and maintain soil water holding capacity, because of high temperature. But this study does not describe about irrigation and soil condition.

Answer: The text related to it has been added to revision.

The pots were soaked with tap water 24 hrs before planting. Soil water holding capacity was not measured as the plants were irrigated (1.5-2 litres of water) daily to avoid any water stress.

Comment-6 Figure 5, picture does not show treatments

Answer: Treatments and detail information’s have been added in the figure.

Comment-7 In discussion section most of references are outdated, need to include recent studies. Moreover, there are many reports on the mitigation of heat stress in plants by AMF and plant physiological changes. What is new findings in this study? Please describe about it as well.

Answer: References has been updated in the manuscript and this study discusses the mechanism of AMF-induced protection of photosynthetic apparatus of soybean against heat stress. Till now, many studies have been conducted on heat stress and AMF separately but to best our knowledge, this is the first elaborative report about alterations in gas exchange parameters and photosynthetic apparatus in response to heat stress in the presence of AMF.

Comment-8 Moreover, soybean is leguminous plant, why no symbiotic performance of plant with rhizobia were described. Also nutrient acquisition of plant, e.g. N,P,K contents in plant tissue should be investigated.

Answer: Before sowing, the seeds were treated with recommended fungicides, viz. bavistin and dithane M and inoculated with slurry of Rhizobium japonicum. Microcosm trial was conducted in Vertisols soil [pH 7.5 (1:2.5 soil water ratio); organic carbon (0.5%); Olsen P (6.2 mg/kg); mineral N (6.4 mg/kg)] with AMF and without AMF colonization. For AMF treated set, the starter culture of 50 g crude soil AM inoculum (mixed native AMF culture, G. intraradices; G. mosseae; G. geosporum) containing roots bits, and hyphal and mycelial mass per pot was inoculated by layering method just below the seeds was inoculated by layering method just below the seeds, completely intermixed in the potting mixture whereas, in case of non-AMF set (control), no starter culture was added. Bradyrhizobial cultures were mixed with sterilized charcoal power (CFU 109/g) and inoculated as seed treatment @ 5 g per kg seed just before sowing. N P and K contents weren’t measured in this study, but still, the study is significant and in future we will consider your suggestion.

Comment-9 Conclusion is too general, what was the hypotheses of this study?, what is new in this study? The conclusion should solely support the objectives of the study.

Answer: Conclusion has been improved

Material and methods

Comment-10 Line 229:

Answer: The pot experiments and greenhouse conditions should be clearly described. From where soil samples were collected, soil chemical physical characterization, irrigation and soil moisture description.

Answer: Many studies have been conducted on AMF and abiotic stress, but all the studies have been conducted in controlled conditions in sterilized soil. Under sterilized soil actual field conditions cannot be mimicked therefore in this study unsterilized or natural soil was used for the experiments. Microcosm trial was conducted in Vertisols soil [pH 7.5 (1:2.5 soil water ratio); organic carbon (0.5%); Olsen P (6.2 mg/kg); mineral N (6.4 mg/kg)]. the plants were irrigated daily to avoid water stress.

Comment-11 Line 289: “total dry weight”, plant or seeds?

Answer: At harvest maturity, plants from the remaining five pots from each treatment were sampled and data on seed weight, pod number, seed number, and total dry weight per plant was recorded.

Round 2

Reviewer 2 Report

I only have a couple of comments regarding the revised manuscript, both related to my original comments. First, though, I'd like to thank the authors for their thoughtful replies and extensive revisions to the manuscript, which generally addressed all of my concerns.

As a result, I don't have any major concerns for another round of revisions. Having said that, I still am concerned about the lack of N and P data - particularly leaf [N] and [P] - as it's difficult to make an argument about how AM fungi affected photosynthetic performance in this study without those data. Along those lines, assigning cause and effect to changes in photosynthetic rates and stomatal opening are difficult, as each could be driving the other - or changes in both may be a side effect of some other factor.

Author Response

We thank the reviewer for providing useful suggestions that have helped us to further improve this manuscript. Following are the responses to the comments of the reviewer. Yes, we agree with this point as AMF colonization is widely believed to stimulate nutrient uptake in plants. Inoculation of AMF can enhance the concentration of various macro-nutrients and micro-nutrients significantly, which leads to increased photosynthate production and hence increased biomass accumulation. As many reports are available on the role of mycorrhiza on plant growth promotion and nutrient management in sustainable agriculture. So, we haven’t measured N and P contents in this study. Though, we have measured other parameters like Nitrogenase activity, leg haemoglobin content, hemechrome and ureids content in root nodules. We have also measured total soluble protein in root nodules and leaves, total chlorophyll content, total free amino acids, and nitrate reductase. Many stomatal and non-stomatal factors are responsible for the enhancement of photosynthetic rate. A significant increase of Photosynthesis in AMF (+) plants suggested that AMF (+) plant's leaves absorbed more solar radiation resulting in enhanced carbon assimilation as compared to control plants which were not colonized with AMF. The transpiration rate also increased AMF (+) plants. Intercellular CO2 was recorded low in AMF (+) plants as compared to control plants indicating that AMF (+) plants had higher photosynthetic performance as compared to AMF (−) in which photosynthetic performance was low. AM symbiosis also increased the number of stomata. Thus it can be said that AM symbiosis offered high gas exchange ability by declining stomatal resistance and by enhancing CO2 assimilation and transpiration fluxes for better photosynthesis.